# A Novel Multiple Role Evaluation Fusion-Based Trust Management Framework in Blockchain-Enabled 6G Network

**DOI:** 10.3390/s23156751

**Published:** 2023-07-28

**Authors:** Yujia Yin, He Fang

**Affiliations:** School of Electronic and Information Engineering, Soochow University, Suzhou 215301, China; 2028410070@stu.suda.edu.cn

**Keywords:** trust evaluation, multiple roles of nodes, blockchain, anomaly detection, neural network, zero trust network access

## Abstract

Six-generation (6G) networks will contain a higher density of users, base stations, and communication equipment, which poses a significant challenge to secure communications and collaborations due to the complex network and environment as well as the number of resource-constraint devices used. Trust evaluation is the basis for secure communications and collaborations, providing an access criterion for interconnecting different nodes. Without a trust evaluation mechanism, the risk of cyberattacks on 6G networks will be greatly increased, which will eventually lead to the failure of network collaboration. For the sake of performing a comprehensive evaluation of nodes, this paper proposes a novel multiple role fusion trust evaluation framework that integrates multiple role fusion trust calculation and blockchain-based trust management. In order to take advantage of fused trust values for trust prediction, a neural network fitting method is utilized in the paper. This work further optimizes the traditional trust management framework and utilizes the optimized model for node trust prediction to better increase the security of communication systems. The results show that multiple role fusion has better stability than a single role evaluation network and better performance in anomaly detection and evaluation accuracy.

## 1. Introduction

An upgraded version of 5G, 6G transmits data and signals via terrestrial wireless devices and satellites, thus expanding the communication range and extending the network to all corners of the globe [1,2,3]. Furthermore, 6G’s satellite network will greatly enhance the transmission capability of connected devices and will be extremely transformative in the fields of natural disaster prediction, satellite positioning, and autonomous driving. However, the increased coverage of the 6G network will cause a proliferation of communication devices, and traditional network solutions authorized by implicit trust relationships will be more vulnerable to cyberattacks [4,5], resulting in the leakage of important and private data [6]. At the same time, with the explosive growth of user service demand, how to utilize limited wireless resources to carry more wireless services has become a challenging issue in 6G networks [7,8].

Zero Trust Network Access (ZTNA) assumes that all connected devices, users, and applications within the network coverage are untrustworthy [9] and performs a real-time authentication and trustworthy evaluation of the requested object before each authorized access [10,11,12]. Each request is confirmed to be legitimate before access is granted to proceed. Trust evaluation is a ZTNA [13,14]. Trust evaluation is performed by an independent group of observation nodes to calculate trust values based on the performance of the observed nodes while performing communication tasks. Once a malicious node launches an attack behavior [15,16,17], the trust value will be anomalous, and the communication system can detect the information about the attack behavior with the trust value [18,19,20].

Trust evaluation can be added to a 6G network to secure the network environment, as it is fair to all nodes in the network, and there is no privileged node. Each request of nodes to access network resources is verified in real time by the trust evaluation system on their trust values to ensure that the node is trustworthy before agreeing to the request. No matter what role the malicious nodes utilize to lurk in the network, they will finally be detected through real-time verification and evaluation. As a result, it is difficult to implement the attack behavior under real-time trust evaluation mechanism.

Trust evaluation has been widely studied in the literature to optimize the network structure and enhance network security. “A jury-based trust management mechanism in distributed cognitive radio networks” proposes a jury-based trust management mechanism. The “jury user” is designed to collaboratively examine the reputation of the cognitive user in the network [21]. This approach utilizes cooperation and mutual supervision for trust management to ensure the accuracy of trust values. “Enhancing Trust Management via Blockchain in Social Internet of Things” and “A blockchain-based vehicle-trust management framework under a crowdsourcing environment” both enhance trust management via blockchain technology. The difference is that “Enhancing Trust Management via Blockchain in Social Internet of Things” utilizes consortium blockchain [22], but “A blockchain-based vehicle-trust management framework under a crowdsourcing environment” uses public blockchain with an efficient consensus algorithm Zyzzyva [23].

By comparison, it was found that using blockchain technology for trust management is an efficient approach. It is possible to accurately record the calculated fusion trust values in real time by modifying the content of the smart contract. The two main features of blockchain are decentralization and data immutability [24,25,26]. The ledger is no longer run by a single center; each node owns a copy of the ledger, and the information in the ledger is not allowed to be tampered with at will. Nodes only have the right to view transactions, not modify them. If blockchain is used in trust management, this can protect the system from a single point of attack to some extent, and also ensure transparency and reliability in trust value management. Otherwise, blockchain is a mature technology that is simpler to operate and can make the process of trust management easier by modifying consensus algorithms [27,28] as well as smart contracts. These recorded trust values can end up being used for anomaly detection as well as trust prediction.

Their models are complete and feasible, but the multiple roles of the nodes in the network are not considered. If a node performs well in a network for a long time and shows high competence, then it will be eligible to assume multiple roles, which ensures trustworthiness and saves network resources. As the network scales up, the trend of multiple roles for nodes is inevitable. Once a node has multiple roles, the single-dimensional trust model will no longer be applicable. As the performance of different roles in performing tasks is biased, it is not possible to generalize the overall trustworthiness of this node by the trust value under only one role.

As a result, we propose a trust fusion evaluation mechanism to improve the evaluation of communication nodes and provide a more comprehensive assessment of node behavior. The main design idea of ZTNA is real-time verification and evaluation. In the system we designed, all network nodes are assumed to be untrustworthy, and their trust level is dynamically generated [29] based on the performance of the communicating nodes in the communication task. The changes in the trust value under each role of a node are fused and thus reflected in the fusion trust value. Any kind of attack committed by virtue of a particular identity is immediately detected in the fusion trust value. Meanwhile, the dynamically generated trust values are managed through blockchain technology, and these recorded trust values can be applied to later anomaly detection and trust prediction, further promoting the development of network security.

The fusion trust values calculated in real time can be used not only as criteria for access, but also for trust prediction. By fitting numerous trust values generated under a node, the trajectory of the trust value can be obtained, so that the behavior of the node can be evaluated more effectively. By analyzing the trend of the trust change curve, potential risks of the system can also be detected to some extent for timely prevention. Neural network fitting can theoretically fit a variety of curves. After training the model, the predicted trust values, and the fitted trust curves, can be computed by simply using the single-dimensional trust values of each role as a feature input. This method has a relatively high prediction accuracy, and neural network fitting is already a mature technique.

In this paper, we aim to provide a more comprehensive and accurate evaluation of the behavior of nodes. The evaluated trust values are also utilized to detect anomalies in the whole system. The security and stability of the network are ensured by these two aspects. The main contributions are summarized as follows:We propose an innovative fusion trust evaluation framework that can conduct a more comprehensive and accurate trust calculation. This framework utilizes blockchain technology to manage the trust value, and makes the management process more transparent and reliable;We develop an algorithm for anomaly detection and a code framework for smart contracts. Anomaly detection provides real-time testing of the system to ensure that the network can operate properly. Smart contract code serves trust management, which makes sure that trust management is carried out properly;We utilize software simulations to verify the feasibility of the proposed framework. Meanwhile, we compare it with a single role evaluation system, and find its superiority in terms of performance. A neural network fitting approach is also applied to trust prediction and compared with conventional linear prediction.

The rest of the paper is organized as follows. The overall architecture of the proposed model is presented in Section 2. Section 3 explains the details of trust calculation and the anomaly detection algorithm. The approach for trust management and the corresponding smart contract code framework are described in Section 4. Section 5 shows the simulation results of this proposed framework. Section 6 is the conclusion of this paper.

## 2. System Design

The whole system is divided into two major parts: real-time fusion trust calculation for multiple roles and trust management based on blockchain technology. As shown in Figure 1, a single node has multiple roles (Role i1, Role i2, …) in a network. Observation nodes will calculate the trust value for all roles of the node.

Once all observation trust values are reached, they are aggregated to generate a fusion trust value. The fusion trust value contains the characteristics of the trust values under each role. Ultimately, these trust values are managed by the blockchain. The figure also demonstrates an attack carried out by a particular role.

If we evaluate only one of the roles, the attack behavior of nodes cannot be detected in time. Even if the attack is detected, it is also very difficult to determine the source of the attack, as a node has many roles. Therefore, multiple role fusion evaluation is very critical. Once the attack has occurred, the trust value under that role will be anomalous. Anomalies in the trust value of any role will be reflected in the fused trust value, so we can quickly detect the attack and which role it was initiated by.

Theoretically, the role of a node can be infinite. Under each different role, the definition of trust and the way of trust calculation are different, so the complexity of analysis will be greatly increased. In order to simplify the analysis, this paper focuses on constructing a network model in which the communication layer and transaction layer collaborate to carry out the analysis of trust fusion evaluation framework.

## 3. The Proposed Blockchain-Based Trust Evaluation Framework

This section focuses on the principle of the multiple role fusion trust evaluation mechanism proposed in this paper. The two layers of the network, the communication layer and the transaction layer, operate collaboratively to form a large network in which the nodes have two roles. A set of independently evaluated node queues dynamically generates trust values based on the completion of nodes performing network tasks. The single-dimensional discrete trust values generated in real time are matched and fused to form fusion trust values, which are recorded and managed through the blockchain.

### 3.1. Trust Calculation

The model consists of two layers: communication layer and transaction layer. The tasks performed by the observed nodes in different layers are not the same. Naturally, trust has different meanings on different layers. Therefore, trust needs to be defined separately and cannot be defined by a single formula. Meanwhile, the trust value of each layer should be fused, so the trust needs to be a dimensionless real number. The trust value defined in this paper needs to satisfy the following conditions:

Trust value ∈[0,1];Trust is a dimensionless number;There exists an inverse relationship between trust and loss of information/data;Each layer of trust is independent.

#### 3.1.1. Communication Layer

The observed nodes forward the message from one user to another. In this process, information will inevitably be lost, and transmission errors may occur. Part of the reason is the effect of the transmission channel, and another part is the error in the forwarding of information by the observed nodes. Message loss, which is caused by channel transmission, is inevitable and ever-present. Due to the continuous development of communication technology, the effect of the channel has become negligible, so the loss of information can be approximated as being caused by the latter. Therefore, the reliability of the observed node is measured based on the message loss rate. The greater the packet loss rate, the lower the trust value will be.

We define a threshold of message loss rate ($R_{loss}$) as θ. The value of θ is 0.4. The whole trust value distribution interval is divided into two parts, the part above θ we define as high-risk interval, and the part below θ is defined as low devotion interval. Nodes falling in the high-risk interval are likely to be malicious nodes (MN), and nodes falling in the low-risk interval are likely to be normal nodes (NN). NN does not lose information intentionally when executing the communication task, so the probability that $R_{loss}$ of NN is distributed at a certain value μ, and we can assume that $R_{loss}$ of NN obeys the Gaussian distribution. MN will intentionally lose a large amount of information, thus hindering the normal execution of the communication task. Therefore, we assume a small interval $\left[\alpha ,\beta \right]$ in which $R_{loss}$ of MN is uniformly distributed. The probability density function of $R_{loss}$ is shown below:(1)$$f(R_{loss})=\left\{\begin{array}{l} \frac{1}{\sqrt{2\pi }\sigma }e^{-\frac{{\left(R_{loss}-\mu \right)}^{2}}{2\sigma^{2}}}\;\;\;\;\;\;\;\;\;\;\;\;\;\;\;\;\;\;\;\;R_{loss}<\theta \\ \frac{1}{\beta -\alpha }\;\;\;\;\;\;\;\;\;\;\;\;\;\;\;\;\;\;\;\;\;\;\;\;\;\;\;\;\;\;\;\;\;\;\;\;R_{loss}\in \left[\alpha ,\beta \right],\;\alpha >\theta_{-},\;\beta <\theta_{-}+\theta_{∆} \end{array}\right.$$
where σ represents variance, and μ is the mean value of $R_{loss}$. $\theta_{-}\to$1, $\theta_{∆}\to$0.

The value of $R_{loss}$ is the ratio of the amount of forwarded messages loss to the amount of original messages:(2)$$\left\{\begin{array}{l} R_{loss}=\frac{-\sum_{i=1}^{N}K_{i}\log_{2}P(\varphi_{i})+\sum_{i=1}^{N}K_{ij}\log_{2}P(\varphi_{i})}{\sum_{i=1}^{N}K_{i}\log_{2}P(\varphi_{i})}\\ \sum_{i=1}^{N}K_{i}=N\\ \sum_{i=1}^{N}P(\varphi_{i})=1 \end{array}\right.$$
assume that the information source is a discrete source consisting of *N* symbols. The number of occurrences of each symbol $\varphi_{i}$ is $K_{i}$, and the probability of occurrence is $P(\varphi_{i})$. $K_{ij}$ represents the number of each symbol left after the forwarding process.

The trust value of the communication layer $T_{C}$ is defined as:(3)$$T_{C}=(1-\frac{\left|R_{loss}-\mu \right|}{\mu })\times sgn(\theta -R_{loss})$$
where $T_{c}$ means the degree of deviation of $R_{loss}$ from μ. The greater the degree of deviation, the worse the node performs the communication task and therefore the lower the trust value will be. Meanwhile, if $R_{loss}<\theta ,$ then the value of trust will be reduced to zero.

#### 3.1.2. Transaction Layer

The definition of trust value at the transaction layer is very similar to the definition of trust value in the communication layer. The probability density function of $R_{loss}$ complies with Equation (1). A transaction data contains $M_{i}$ indicators, such as item name, price, transaction number, transaction status, etc. Different transactions have different indicators. The number of data generated per transaction is *N*. Then, the total amount of data for the transaction $D_{0}$ is:(4)$$D_{0}=\sum_{i=1}^{N}M_{i}$$

The loss rate of data $R_{dloss}$ is the ratio of the amount of data loss to the amount of original data:(5)$$R_{dloss}=\frac{D_{0}-\sum_{i=1}^{N}M_{ij}}{D_{0}}$$
where $M_{ij}$ is the data recorded from $D_{0}$.

The trust value of the transaction layer $T_{T}$ is:(6)$$T_{T}=(1-\frac{\left|R_{dloss}-\mu_{T}\right|}{\mu_{T}})\times sgn(\theta -R_{dloss})$$

#### 3.1.3. Fusion Trust

The fusion trust value must be able to reflect the change in trust at both the communication and transaction layers. That means the communication layer and transaction layer trust values become the two characteristics of the fused trust value:(7)$$T=f(T_{C},T_{T})$$

To simplify the analysis, we assume that the three variables obey a linear relationship:(8)$$\left\{\begin{array}{l} T=aT_{C}+bT_{T}\\ a+b=1 \end{array}\right.$$
where a and b are weighted values of two single-layer trust values. The sum of a and b is 1, thus ensuring that the fusion trust value ranges between 0 and 1.

### 3.2. Fusion-Based Trust Anomaly Detection Algorithm

This section explains how the fusion trust evaluation proposed in this paper can be used for trust anomaly detection to detect the occurrence of an attack and the source of the attack on time. The notations used in anomaly detection algorithm and their corresponding meanings are shown in Table 1.

We use the amount of change in the trust value (∆T) as an indicator for anomaly detection. When NN performs a normal network task, the trust value fluctuates up and down in small increments around a fixed value ($\mu_{*}$). If the change in trust value is very drastic and exceeds a certain acceptable threshold (ρ), then an anomaly can be considered to have occurred. According to the formula presented in Section 3.1, we can obtain ∆T in the abnormal state:(9)$$\left\{\begin{array}{l} a(1-\frac{\left|R_{loss}-\mu \right|}{\mu })+b\frac{\left|r_{dloss}-\mu_{T}\right|-\left|R_{dloss}-\mu_{T}\right|}{\mu_{T}}\;\;\;\;\;\;\;\;\;\;\;\;\;\;\;\;\;\;\;\;\;{\;\;\;R}_{loss}\in \Omega_{HR},\;R_{dloss}\in \Omega_{LR}\\ b(1-\frac{\left|R_{dloss}-\mu \right|}{\mu })+a\frac{\left|r_{loss}-\mu_{T}\right|-\left|R_{loss}-\mu_{T}\right|}{\mu_{T}}\;\;\;\;\;\;\;\;\;\;\;\;\;\;\;\;\;\;\;\;\;\;\;\;\;R_{dloss}\in \Omega_{HR},\;R_{loss}\in \Omega_{LR}\\ a(1-\frac{\left|R_{loss}-\mu \right|}{\mu })+b(1-\frac{\left|R_{dloss}-\mu \right|}{\mu })\;\;\;\;\;\;\;\;\;\;\;\;\;\;\;\;\;\;\;\;\;\;\;\;\;\;\;\;\;\;\;R_{loss}\in \Omega_{HR},\;R_{dloss}\in \Omega_{HR}\;\\ a\frac{\left|r_{loss}-\mu_{T}\right|-\left|R_{loss}-\mu_{T}\right|}{\mu_{T}}+b\frac{\left|r_{dloss}-\mu_{T}\right|-\left|R_{dloss}-\mu_{T}\right|}{\mu_{T}}\;\;\;\;\;\;\;\;\;\;\;\;\;\;\;\;\;\;\;\;\;\;\;R_{loss}\in \Omega_{LR},\;R_{dloss}\in \Omega_{LR} \end{array}\right.$$
after calculating the value of ∆T, we just need to compare this value with ρ to evaluate the behavior of the node.

The source of the anomaly can be found by comparing the value obtained by $\frac{\partial T}{\partial a}-\frac{\partial ∆T}{\partial a}$ and $\frac{\partial T}{\partial b}-\frac{\partial ∆T}{\partial b}$. The value of $\frac{\partial T}{\partial a}-\frac{\partial ∆T}{\partial a}$ and $\frac{\partial T}{\partial b}-\frac{\partial ∆T}{\partial b}$ are shown in Table 2. As long as the value is 0, it means that there is an anomaly in that layer. × means that the result is an arbitrary real number that is not 0.

The core idea of anomaly detection algorithm in this model is to loop through the trust value of the previous iteration and the adjacent trust value to determine the state of the difference. Algorithm 1 shows the fusion-based trust anomaly detection algorithm.
**Algorithm 1: Fusion-based trust anomaly detection algorithm.****Input:**ζi, ρ, μ***Output:**ηti, ϱSR, ςt, S2**begin**1:    detection system is ready 2:    **return**ηti3:    **for** *n* = 1: length(ζi) 4:            Tn=ζi(n), Tn+1=ζi(n+1)5:            ∆T=Tn−Tn+16:            **if**
∆T<ρ
**then**7:                be considered normal8:            **else**9:                Tn+1 is added to ξt(n)10:      **end for**11:      **foreach**
*T* in ξt(n)12:         **if**
∂Tn∂a−∂∆T∂a=0, ∂Tn∂b−∂∆T∂b≠013:               anomaly source is in communication layer14:         **elseif**
∂Tn∂a−∂∆T∂a≠0, ∂Tn∂b−∂∆T∂b=0
15:               anomaly source is in transaction layer16:         **elseif**
$\frac{\partial T_{n}}{\partial a}-\frac{\partial ∆T}{\partial a}=0,\;\frac{\partial T_{n}}{\partial b}-\frac{\partial ∆T}{\partial b}=0$
17:               anomaly source is in both layers18:         **return**
ϱSR19:      **end for**20:      S2=1n−1∑i=1n(NTi−μ*)221:      **return**
S2, ςt**end**

The steps involved in the algorithm are shown below:

Step 1:Get fusion trust values $T_{n}$ and $T_{n+1}$ from $\zeta_{i}$. After the acquisition is complete, to ensure repeat detection, it needs to be marked with $\eta_{t}^{i}$ and returned to the trust manager to indicate that the fusion trust value at that time has been received for detection.Step 2:Calculate $∆T=T_{n}-T_{n+1}$. The value of ∆T is compared with ρ. If ∆T is less than the threshold, it can be considered normal. If the value is greater than the threshold, it can be considered to have a high probability of an abnormal condition. The filtered trust values with a high probability of anomalies are stored in a list $\xi_{t}$.Step 3:Calculate $\frac{\partial T_{n}}{\partial a}-\frac{\partial ∆T}{\partial a}$ and $\frac{\partial T_{n}}{\partial b}-\frac{\partial ∆T}{\partial b}$. If the value is 0, then it is assumed that an abnormal condition has occurred at that layer. After all the fusion trust values have been detected, a list $\varrho_{S}^{R}$ storing the detection results is returned.Step 4:Calculate the deviation of all trust values ($S^{2}$) judged as normal from the theoretical trust value of the system. The value of $S^{2}$ provides a comprehensive measure of the stability of nodes performing communication tasks.Step 5:When the exception detection is complete, an end flag $\varsigma_{t}$ is returned. All stored lists ($\zeta_{i}$, $\xi_{t}$, $\varrho_{S}^{R}$) will be cleared for the next round of detection.

## 4. Blockchain-Based Trust Management

In this section, we present the trust management part of the fusion trust evaluation model proposed in this paper. Trust management is implemented mainly by using a smart contract on the blockchain. The trust values recorded in the block are managed by modifying the smart contract.

We chose consortium blockchain as a tool for trust management due to its advantages in data storage. Consortium blockchain is jointly maintained by multiple organizations involved, and they share resources and responsibilities to ensure the stability and reliability of the whole system. In the consortium blockchain, smart contracts perform various operations automatically and transparently, and each participant can view and verify the data at any time. This transparency increases the trustworthiness of the entire system and reduces the risk of fraud and manipulation. If the consortium blockchain [30,31] is applied to the model proposed in this paper, then it can make the record of the trust value of each round more accurate and ensure the openness and transparency of the fusion trust record.

Hyperledger Fabric is a consortium blockchain [32,33]. The sample network of Fabric is shown in Figure 2. This network is constructed by an organization of R1, R2, and R3. The configuration of the network is preserved in C1. O is a sorting service node that was first defined in the sample network. CA1, CA2, and CA3 are Certificate Authorities belonging to R1, R2, and R3, respectively. Certificate Authorities (CA) [34] assign certificate X.509, which can be used to identify components belonging to the organizations R1, R2, and R3. The alliances R1, R2, and R3 have formed to add a channel to this network. Peers P1 and P2 are fundamental elements of the network, carrying copies of ledgers L1, L2, and chaincode (contains smart contract S1, S2). With this channel, the App can access the ledger by invoking chaincode. By understanding the architecture of the entire network, it is possible to understand how the chaincode functions in the network. First, we need to write a smart contract [35] as an organizer of the network. Then, package the smart contracts in the form of chaincode, and install the package on peers. Next, only when all organizations in the alliance have approved the chaincode definition can the chaincode be committed to the channel. Finally, the App can invoke chaincode to have access to the context of ledger. Most of the steps can be done on the fabric platform by calling the relevant commands directly.

Trust management mainly uses smart contracts to invoke the ledger and record the fusion trust values into the ledger. When the fusion trust values need to be retrieved for data processing, the smart contract also needs to be invoked to get the ledger information. The details of the smart contract are shown in Algorithm 2.
**Algorithm 2: Trust Management Based on Smart Contract.****Input:** Fusion trust
**Output:** Ledger**begin**1:         **%** InitTrust: Initialize the ledger and assign initial values to the trust values.2:         define a structure Trust containing *N*,id1,id2,t,trust1,trust2
3:         Assign an initial value to each structure element4:         **%** CreateTrust: Set a new trust record. 5:         Define a newly written trust value6:         Check if the trust value is repeated7:         **if** not repeated **then**8:             **return** ctx.GetStub().PutState(*N*,id1,id2,t,trust1,trust2)9:         **end if**10:       % ReadTrust: Read a trust record.11:       ctx.GetStub().GetState(*N*,id1,id2,t,trust1,trust2)12:       % UpdateTrust: Update the trust value.13:       Change the value of trust14:       Check if the trust value is repeated15:       **if** not repeated **then**16:           **return** ctx.GetStub().PutState(*N*,id1,id2,t,trust1,trust2)17:       **end if**18:    **%** DeleteTrust: Delete a trust record.19:    **return** ctx.GetStub().DelState(*N*,id1,id2,t,trust1,trust2)**end**

Within Hyperledger Fabric, a high-level Application Programming Interface (API) is provided. When using the contract API, each chaincode function that is called is passed a transaction context “ctx”, from which you can get the chaincode stub (GetStub()), which has functions to access the ledger (e.g., GetState()) and make requests to update the ledger (e.g., PutState()).

In Hyperledger Fabric, the chaincode can only function if it is deployed to the channel. The steps for deploying a chaincode are shown below:Package the smart contract. Package smart contract into chaincode before it can be installed on peers;Install the chaincode package. After packaging the smart contract, we can install the chaincode on our peers. The chaincode needs to be installed on every peer that will endorse a transaction;Approve a chaincode definition. After installing the chaincode package, we need to approve a chaincode definition for the organizations. The definition includes the important parameters of chaincode governance such as the name, version, and the chaincode endorsement policy;Commit the chaincode definition to the channel. After a sufficient number of organizations have approved a chaincode definition, one organization can commit the chaincode definition to the channel. If a majority of channel members have approved the definition, the commit transaction will be successful and the parameters agreed to in the chaincode definition will be implemented on the channel;Invoke the chaincode. After the chaincode definition has been committed to a channel, the chaincode will start on the peers joined to the channel where the chaincode was installed. The chaincode is now ready to be invoked by client applications.

## 5. Results

This section will show the superiority of the proposed scheme in this paper by comparing it with traditional trust management models, and show how neural networks achieve trust prediction compared to linear weighted trust prediction.

### 5.1. Performance of Fusion Trust Evaluation

The accuracy of attack detection is defined as the ratio of the number of attacks detected by the system to the total number of attacks. The accuracy of the two-dimensional fusion trust evaluation network in detecting attack behavior is compared with that of the single dimension trust evaluation network. As shown in Figure 3, with the increasing evaluation process, the prediction rate of the two-dimension fusion network is stable at around 0.725, which is higher than either the transaction or communication layers. The accuracy of attack prediction for communication layer and transaction layer single-dimension trust is very close.

It is possible that a malicious node with multiple roles will not use all of them to carry out an attack, but only one role to implement the attack. The layer of the network that is not under attack will naturally consider the node a good node, but the two-dimension fusion network captures the behavior of the node under all roles.

If the observed node is good, its behavior will be relatively stable in the absence of accidents. Therefore, the trust value will fluctuate up and down around a fixed value. We test the degree of fluctuation of the trust value calculated by the two-dimension fusion network and the single dimension network with respect to a fixed value to measure the stability of the two kinds of systems. As shown in Figure 4, the volatility of a two-dimension fusion network is smaller than that of a single dimension network, which means the multiple role fusion framework is more stable.

With the increasing evaluation process, two-dimension fusion networks tend to grow more flatly in volatility than single dimension networks. The communication layer and transaction layer have almost the same fluctuation curve.

### 5.2. Performance of Trust Prediction

We used 500 data points for the training of the neural network. The trust value of the communication layer and the trust value of the transaction layer were two dimensions of the input data, respectively. The 500 data points were divided in a 5:1:1 ratio into training data, validation data, and test data. The hidden neurons were 10. The error histogram is shown in Figure 5. The error distribution was between −0.08021 and 0.1005.

Figure 6 shows the Mean Square Error (MSE) after the neural network prediction and the linear network prediction. MSE measures the extent to which the predicted value matches the true value. MSE is calculated using the formula $\frac{1}{n}\sum_{i=1}^{n}{\left(T_{i}-{\hat{T}}_{i}\right)}^{2}$. $T_{i}$ is the real trust value, ${\hat{T}}_{i}$ is the predicted value. We calculated the MSE of the neural network prediction as well as linear network prediction. It can be seen that the MSE of the neural network prediction was significantly smaller than that of the linear network. That is, the trust value predicted by the neural network was closer to the true trust value of the nodes.

## 6. Conclusions

In this paper, we proposed a multiple role evaluation fusion-based trust management framework for blockchain-enabled wireless communication system, which is in line with the ZTNA design philosophy and can be applied to improve the security of 6G. This framework performs a comprehensive evaluation of the nodes, uses blockchain for trust value management, and finally uses neural network fitting for trust value prediction. Compared with the traditional model, we made three optimizations to the single-dimension trust evaluation model, aiming to increase the security and reliability of communication and also make the common trust mechanism better.

## Figures and Tables

**Figure 1 sensors-23-06751-f001:**
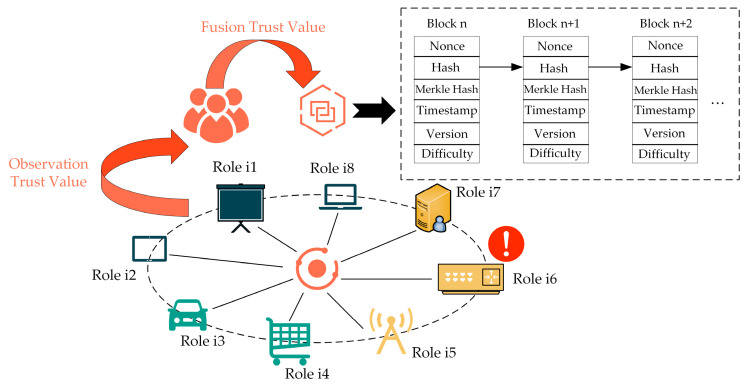
The proposed multiple role fusion-based evaluation and blockchain-enabled trust management framework.

**Figure 2 sensors-23-06751-f002:**
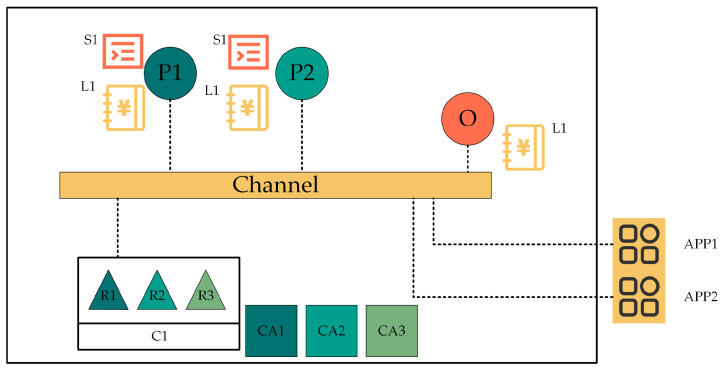
A sample network structure of Hyperledger Fabric.

**Figure 3 sensors-23-06751-f003:**
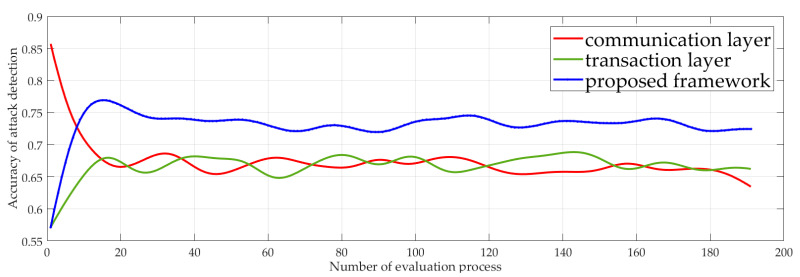
The accuracy of the proposed model and the traditional one-dimensional trust evaluation model in detecting malicious behavior.

**Figure 4 sensors-23-06751-f004:**
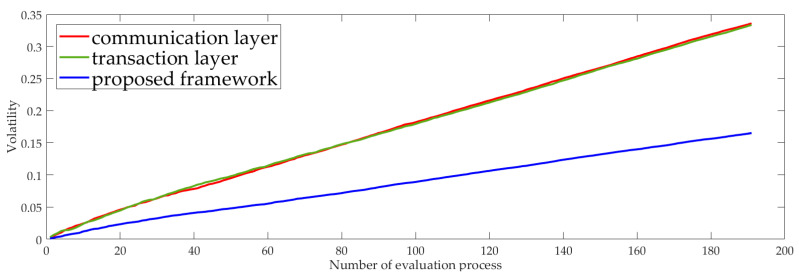
The fluctuation of the proposed model and one-dimensional network between the actual value and the true value reflects the stability level of the system.

**Figure 5 sensors-23-06751-f005:**
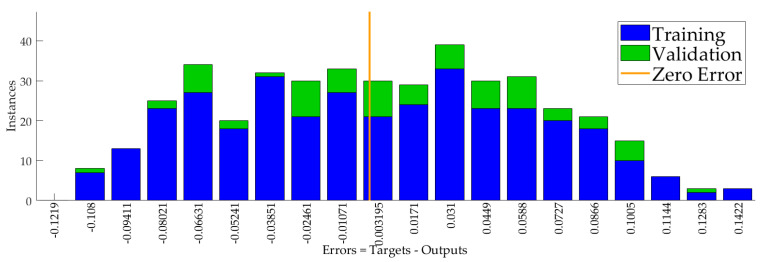
The error histogram of neural network shows the error distribution of training and validation data.

**Figure 6 sensors-23-06751-f006:**
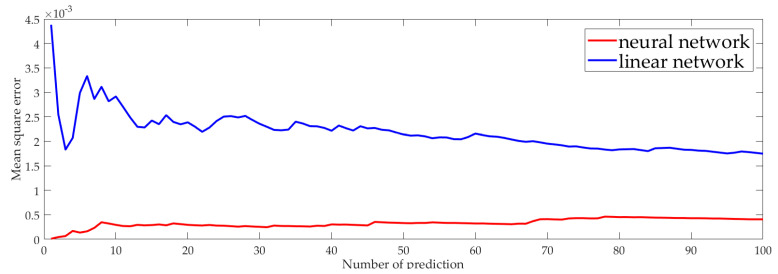
The Mean Square Error of neural network and linear network.

**Table 1 sensors-23-06751-t001:** Notation table.

Notation	Meaning
∆T	The amount of change in the trust value.
$\mu_{*}$	A fixed value of trust.
ρ	Trust threshold for anomaly detection.
$\Omega_{HR}$	The high-risk interval.
$\Omega_{LR}$	The low-risk interval.
$T_{n}$	The fusion trust value at the *n*th iteration.
$\zeta_{i}$	List of stored fusion trust values tagged with serial number *i*.
$\eta_{t}^{i}$	The token that needs to be returned is tagged with detection time *t.*
$\xi_{t}$	List of stored abnormal fusion trust values tagged with detection time *t.*
$\varrho_{S}^{R}$	List of detection results. R: abnormal or not, S: source of anomaly.
$S^{2}$	Measuring the degree of deviation between normal trust value and $\mu_{*}$.
$\varsigma_{t}$	Status flag for the end of anomaly detection.
*NT*	Fusion trust values which are determined to be normal.

**Table 2 sensors-23-06751-t002:** The value of $\frac{\partial T}{\partial a}-\frac{\partial ∆T}{\partial a}$ and $\frac{\partial T}{\partial b}-\frac{\partial ∆T}{\partial b}$.

$$\frac{\partial T}{\partial a}-\frac{\partial ∆T}{\partial a}$$	$$\frac{\partial T}{\partial b}-\frac{\partial ∆T}{\partial b}$$	Conclusion
0	0	Both layers are abnormal.
0	×	Communication layer is abnormal.
×	0	Transaction layer is abnormal.
×	×	Both layers are normal.

## Data Availability

Our data is generated by software, therefore we are unable to provide specific data.

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
