# Peer review of "A Novel Multiple Role Evaluation Fusion-Based Trust Management Framework in Blockchain-Enabled 6G Network"

_sensors, 2023, doi:10.3390/s23156751_

Round 1
Reviewer 1 Report
In this paper, the authors studied a multiple role evaluation fusion-based Trust management framework in blockchain-enabled 6G networks. However, I have some comments.
1. Please perform thorough revision to remove any grammatical errors.
2. Literature review is incomplete. Latest works in technical literature which study efficient resource allocation in futuristic 6G networks should be added such as
[a] Ranjha, A., Javed, M.A., Srivastava, G. and Lin, J.C.W., 2022. Intercell Interference Coordination for UAV enabled URLLC with perfect/imperfect CSI using cognitive radio. IEEE Open Journal of the Communications Society.
[b] Asif, M., Ihsan, A., Khan, W.U., Zhang, S. and Wu, S.X., 2022. Energy-efficient backscatter-assisted coded cooperative noma for b5g wireless communications. IEEE Transactions on Green Communications and Networking, 7(1), pp.70-83.
3. Why is trust evaluation important for secure communications and collaborations in 6G networks?
4. How can a novel multiple role fusion trust evaluation framework contribute to enhancing the security of communication systems in 6G networks?
5. What role does blockchain-based trust management play in the proposed trust evaluation framework?
6. How does the neural network fitting method contribute to utilizing fused trust values for trust prediction?
7. According to the results, what advantages does the multiple role fusion have over a single role evaluation network in terms of stability, anomaly detection, and evaluation accuracy?
Please perform thorough revision to remove any grammatical errors.
Reviewer 2 Report
A novel multiple role evaluation fusion-based trust management framework has been proposed in this paper. It utilizes the blockchain to enhances the transparency and reliability of trust management. A detailed theoretical description as well as the necessary simulation verification have been developed to demonstrate the feasibility of this proposed framework. I still has some comments:
1. Equation (8) does not seem to be explained clearly enough. The parameters in the equation need to be explained in more detail to ensure that the reader can grasp the meaning of the equation.
2. Algorithm 2 does not provide enough details, for example, it does not explain some of the functions involved in the algorithm. This may be confusing.
3. The text does not provide a detailed explanation of exactly how the MSE in Figure 6 is calculated. This may not be conducive for readers to reproduce the simulation result.
The english can be improved.
Round 2
Reviewer 1 Report
All comments are addressed.